# Pillar Growth by Focused Electron Beam-Induced Deposition Using a Bimetallic Precursor as Model System: High-Energy Fragmentation vs. Low-Energy Decomposition

**DOI:** 10.3390/nano13212907

**Published:** 2023-11-06

**Authors:** Robert Winkler, Michele Brugger-Hatzl, Fabrizio Porrati, David Kuhness, Thomas Mairhofer, Lukas M. Seewald, Gerald Kothleitner, Michael Huth, Harald Plank, Sven Barth

**Affiliations:** 1Christian Doppler Laboratory—DEFINE, Graz University of Technology, 8010 Graz, Austria; 2Graz Centre for Electron Microscopy, 8010 Graz, Austria; 3Institute of Physics, Goethe University, Max-von-Laue-Str. 1, 60438 Frankfurt, Germanymichael.huth@physik.uni-frankfurt.de (M.H.); 4Institute of Electron Microscopy, Graz University of Technology, 8010 Graz, Austria; 5Institute for Inorganic and Analytical Chemistry, Goethe University Frankfurt, Max-von-Laue-Str. 7, 60438 Frankfurt, Germany

**Keywords:** additive direct-write manufacturing, 3D nano printing, focused electron beam-induced deposition, nanomagnetic, magnetic force microscopy, chemical composition

## Abstract

Electron-induced fragmentation of the HFeCo_3_(CO)_12_ precursor allows direct-write fabrication of 3D nanostructures with metallic contents of up to >95 at %. While microstructure and composition determine the physical and functional properties of focused electron beam-induced deposits, they also provide fundamental insights into the decomposition process of precursors, as elaborated in this study based on EDX and TEM. The results provide solid information suggesting that different dominant fragmentation channels are active in single-spot growth processes for pillar formation. The use of the single source precursor provides a unique insight into high- and low-energy fragmentation channels being active in the same deposit formation process.

## 1. Introduction

Focused electron beam-induced deposition (FEBID) is a well-established method for the direct writing of nanoscale materials and has emerged as a viable technique for 3D nanoprinting of even complex architectures in a single-step procedure [1]. Additive manufacturing methods have reached a high level of sophistication for objects down to the lower micrometer range, but it is still very challenging when aiming to produce freestanding 3D structures at the sub-micron and, in particular, at the nanoscale [2,3]. The latter follows the general trend of miniaturization of devices and functional 1D–3D structures, which enables the development of novel applications due to functionalities emerging at the nanoscale (e.g., plasmonics, magnetic phenomena) [4,5,6,7,8,9,10,11,12,13,14].

In direct-write processes, pillar growth yields the fundamental building block for the construction of meshed 3D objects, as demonstrated in literature [1]. In this context, a distinction should be made between the intermittent (interlacing, parallel) patterning [15] and the continuous writing of a single wire [16,17]. For the latter, electron beam heating has been identified as a major contributor to precursor depletion due to increased desorption during growth [18], which is accompanied by decreasing growth rates with increasing structure height/length [19,20]. For the final shape and cross-sectional evolution in the as-deposited FEBID wires, the direction of the gas injection system [21]. shadowing effects [22] and beam parameters [19] have to be considered and carefully tuned to achieve the highest shape fidelities. Furthermore, the influence of beam currents on crystallinity and purity has been described. For the formation of 3D nanostructures, the beam current is of particular importance since higher beam currents lead to larger pillar diameters and growth artifacts [21,23]. In addition to the beam parameters affecting the deposition, FEBID relies strongly on the supply of volatile precursors, while the material composition depends on the suitability of the precursors, e.g., their tendency for dissociation upon electron impact [24,25]. Homometallic carbonyls are in particular excellent model systems, allowing side reactions with background gases to be predicted due to their simple stoichiometry and fixed C:O ratio. In addition, metal carbonyls provide access to very high metal contents of more than 95 at %, as comprehensively summarized in recent literature [24].

Bimetallic FEBID materials can be prepared by using suitable single-source precursors [26,27] or by using multiple gas injection systems [28]. Advantages of the single-source approach are the simplicity of precursor adsorption with a predefined metal ratio and the simpler requirements for the injection systems needed for the deposition of material. Moreover, these molecular sources can provide a unique opportunity to study different fragmentation channels in FEBID, as described herein.

Recently, we have published results concerning the direct writing of high-performance bimetallic tips for magnetic force microscopy (MFM) applications [29], revealing superior performance compared to commercial MFM tips [30]. Here, we discuss the microstructural features of HFeCo_3_(CO)_12_-derived nanocones/pillars and relate the observed microstructure as well as chemical composition to the beam conditions. The results suggest a significant influence of the electron energy on the precursor fragmentation path for this anisotropic nanostructure growth.

## 2. Materials and Methods

To generalize the results, FEBID experiments were performed on three focused ion beam (FIB)/scanning electron microscopes (SEM) dual beam microscopes (Nova NanoLab 600; Nova 200; Quanta 3D FEG, Thermo Fisher Scientific, Eindhoven, The Netherlands). For defined HFeCo_3_(CO)_12_ precursor flux, the distance between nozzle and sample surface was set to 100 μm, and the precursor crucible was heated to 55–65 °C. Concerning the synthesis of the Co-Fe precursor, we refer to recent literature [26]. Deposition was carried out at primary electron energies of 5–30 keV and beam currents in the range of 7–130 pA. Pillar structures referred to as ‘cones’ represent an exposure strategy with stepwise decreasing beam blur values as reported in [29], ‘focus pillars’ denote nanostructures fabricated under static in-focus conditions. Electron beam curing (EBC) [4] was carried out for 30 min at 30 keV and 20 nA in top view on a 400 nm wide circular area.

For transmission electron microscopy (TEM) investigations, FEBID pillars have been deposited directly onto FIB-structured OmniProbe copper-based lift-out grids. TEM characterisation was carried out on two microscopes. First, a probe-corrected FEI Titan3 G2 microscope (FEI Thermo Fisher Scientific, The Netherlands) was used and operated at 300 kV. STEM was performed with a high-angle annular dark field detector (HAADF) and evaluated using DigitalMicrograph software and Velox (version 3.0) by Thermo Scientific (Waltham, MA, USA). TEM-based energy dispersive X-ray (EDX) spectroscopy was performed with a high-sensitivity four-quadrant SDD X-ray spectrometer (Super-X, Chemi-STEM technology) using the Gatan Microscopy Suite (version 3.4; Gatan, Inc., Pleasanton, CA, USA) and Velox (version 3.0) by Thermo Scientific (Waltham, MA, USA). For pre-evaluation and double-checking, we used a monochromated Tecnai F20 (Thermo Fisher Scientific, Eindhoven, The Netherlands) operated at 200 keV.

## 3. Results and Discussion

The HFeCo_3_(CO)_12_ precursor has been described to be very reliable for high purity FEBID-based deposition for planar [26] and mesh-like 3D nanostructures [10]. In all these experiments, the Co:Fe ratios have been described to reflect the metal composition in the molecular precursor, while overall purities depended on the microscope background pressure and the applied deposition strategy [10,26]. These results are in good agreement with surface science studies using low-energy electrons, which revealed that pure Co3Fe forms by a thermally supported electron-induced fragmentation process [31]. According to gas-phase and surfaces science studies, the electron-induced dissociation of the HFeCo_3_(CO)_12_ precursor leading to a high CO abstraction should be dominated by dissociative ionisation (DI) [31,32].

In contrast to the writing of meshed objects or planar deposits, vertical pillar growth relies on stationary electron beam exposure. Therefore, the high-energy electron beam is permanently located in the centre of the deposit. In this context, it should be noted that the pillar width is larger than the beam diameter and surface irregularities may occur depending on precursor flux and diffusion phenomena, leading to slight variations [4].

Figure 1a shows high angle annular dark field (HAADF) images of typical HFeCo_3_(CO)_12_-derived nanopillars/cones. In addition to the brightness variation caused by the different thicknesses of the circular cross-sections, all of the prepared nanopillars reveal a darker section in the centre of the deposits. This core channel is observed under both beam conditions, focused (pillars) and blurred (cones) [4], as well as under all beam parameters (5–30 keV/7–129 pA), as shown in Appendix A of the Appendix A. The HAADF images suggest a compositional variation due to the Z-contrast in the centre compared to the rest of the deposited material. The higher contrast is related to the incorporation of lighter elements, such as carbon, as evident by the EDX elemental map of Figure 1b. Since Fe and Co are neighbours in the periodic table, the Z-contrast is neither related to variations in their composition nor to diffraction contrasts due to the generally high crystallinity of the material, as discussed below.

The elemental maps of the tip region reveal an enrichment of Fe and a lower concentration of Co in the centre of the nanostructure. This means that the Co:Fe ratio of deposits made from the HFeCo_3_(CO)_12_ precursor deviates from the expected 3:1 value. The surface plot in Figure 2a illustrates the compositional variation along the nanowire (NW) axis from the tip area with the Fe-dominated EDX signal. The iron-dominated core along the growth axis shows an increasing Co content due to the formation of a shell containing higher cobalt content. Once the shell is thick enough that it can be separated from the core on the EDX map, a constant 3:1 ratio resembling the HFeCo_3_(CO)_12_ precursor is obtained. This levelling to a constant composition is even more evident when comparing the cross-sectional EDX line scans for Co and Fe, as shown in Figure 2b and Appendix A in the Appendix A. The Co:Fe ratio of the shell is consistent with the typical FEBID material derived by in-plane serpentine patterning and wire-frame structure generation [10,26], where the dissociative electron attachment (DEA) channel dominates the fragmentation process, as growth is typically carried out in the precursor limited regime [33].

Consequently, the iron-rich core must be dominated by a different fragmentation channel and an associated preferential decomposition of an iron-rich intermediate. Moreover, the weak C enrichment in the core could be an additional indication of the dissociative ionisation (DI) channel being dominating, as FeC and CoC fragments have been observed before for decomposition at higher electron energies [34,35,36]. Similar results have been reported for monometallic Fe- and Co-based FEBID pillars and nanowires characterised by atom probe tomography [28]. The formation of iron-rich carbonyl fragments could be caused by decomposition of the tetrahedral metal core of HFeCo_3_(CO)_12_, which is consistent with the observed possibility of monometallic iron carbonyl fragment formation by dissociative electron attachment observed in single electron collision studies using molecular beams [33] and cobalt carbonyl abstraction in channels assigned to dissociative ionisation [32]. Similarly, H_2_FeRu_3_(CO)_13_, which is a related precursor, loses some of the iron during FEBID writing, which could be caused by the formation of an iron carbonyl species [32]. Thermal contributions are less likely to be a major contributor since the FEBID-derived high-purity metal deposit is highly heat conducting; the cone is already formed at the base of the nanostructure, and at the same time, stoichiometric deposits are obtained in purely thermally induced processes [9]. Similarly, pillar growth with simultaneous precursor feeds of Co_2_(CO)_8_ and Fe_2_(CO)_9_ shows core-shell formation of a simplified composition of Co_0.55_Fe_0.30_O_0.15_ for the core and a Co_0.95_O_0.05_ shell [28], supporting our hypothesis that Fe-based carbonyls are more receptive to high energy electrons, while SE-induced fragmentation is efficient for stoichiometric deposition of high purity heterometallic Co_3_Fe.

In FEBID, the effective fragmentation yield for a given precursor is a convolution of the spatial electron energy distribution, determined by i.e., primary (PE), secondary (SE), backscattered (BSE), and forward scattered (FSE) electrons together with inelastically scattered electrons, and the energy-dependent cross sections for respective electron-induced processes [36,37]. In general, the vertical growth mode is governed by PE and SE, while the lateral growth is dominated by BSE- and FSE-related SEs [38,39]. Electron trajectory simulations for the specific pillar geometry (cylindrical shaft, conical tip with tip radii in the range of 10 nm [29]) indicate a high density of high-energy electrons at the apex (Appendix A). In agreement with the instrument specification, the beam diameter and PE-related SEs are in the range of the inner Fe-rich core of ~20–70 nm [21]. In a precursor-limited regime, the precursor cross sections are not necessarily the dominant parameter but rather the number of electrons in certain energy ranges. At such exposed regions (sharp tip), the balance of high-energy electrons to secondary electrons shifts in favour of the former, making electron-induced reactions of high-energy electrons with the precursor more likely. Once a higher number of secondary low-energy electrons are available, the relatively high cross-section of HFeCo_3_(CO)_12_ for low-energy DEA and DI or simply their abundance will dominate the fragmentation. Consequently, the pillar will grow laterally to a constant diameter related to the availability of decomposition-inducing electrons and their penetration depth. Hence, the pillar will maintain a constant diameter for a single-spot growth strategy, as illustrated in Figure 1a (right inset). The spatial differences in the ratio of high/low energy electrons result in a variation of the chemical composition, as shown in Figure 1. To the best of our knowledge, such a behaviour has not been observed for FEBID structures before and illustrates that besides optimized growth parameters for specific material deposition, applying heterometallic or generally more complex precursors can provide unique insight into reaction channels not being observed or noticed in single metal deposition studies.

As shown in Figure 3, already as-deposited structures are very crystalline, which is a rare observation for such low-beam currents. A tendency towards higher crystallinity with a dramatic increase in beam currents up to the µA range [21] and elevated substrate temperature [40] has been described in literature. However, the only reported FEBID materials with high crystallinity grown at room temperature and currents in the pA up to low nA ranges are iron-based deposits crystallising predominantly in the α-Fe phase [41,42]. The complexity of a phase identification for the heterobimetallic Co3Fe can be understood considering as-grown homometallic FEBID Co pillar with a combination of face-centred cubic (fcc) and hexagonal close-packed (hcp) Co phases [43]. Besides the thermodynamically stable phases mentioned above, such as fcc and hcp Co, fcc and bcc Fe, as well as fcc and bcc CoFe, other binary phases have been described in the literature. The binary phases include Co_3_Fe, Fe_3_Co, and Fe_9_Co_7_, with cubic and non-cubic unit cells of different dimensions as well as non-ordered substitutional solid solutions or superstructures [44]. Hence, it is not surprising that individual crystal phases cannot be distinguished from fast-Fourier-transforms (FFT). However, as shown by the FFT in Figure 3e,f and Appendix A of the Appendix A, the crystallinity of the as-deposited material is evident. Moreover, post-growth EBC triggers further grain growth with domains up to approx. 20 nm and sometimes, as shown in the HAADF image in Figure 3d. It should be mentioned that some diffraction spots/rings are partly missing or appearing, depending on the FFT region (compare the inner circle in Figure 3e,f). For instance, 0.52 nm d-spacing as well as 0.45 nm d-values are only sometimes observed. Due to the complex nature of the crystalline phases and the composition variation of the inner core, we cannot reliably assign any of the spots to specific phases. However, we would like to point out that even TEM imaging, which can be considered post-growth curing, provides a highly crystalline material, as shown in Figure 3b. Such complete crystallization can be traced back to the easy formation of crystalline phases of a highly pure deposit without the formation of a composite, as observed in most other FEBID materials.

## 4. Conclusions

The observed results illustrate the unique opportunities using single-source bimetallic precursors to observe different fragmentation channels during the actual FEBID process. For instance, the single-spot continuous writing allows us to observe FEBID structures being dominated by different growth regimes, such as high-energy fragmentation channels, and generally low-energy decomposition regimes, such as DEA or DI. The data suggest that Fe-dominated fragmentation of the bimetallic precursor HFeCo_3_(CO)_12_ governs the vertical growth mode of nanopillars due to an altered decomposition path when the high-energy electrons interact with the precursor. EDX reveals an iron-rich core with more than 50 at % Fe, which differs significantly from the 25 at % in the precursor. In contrast, lateral growth is dominated by low-energy electrons, and the deposit maintains the metal stoichiometry provided by the single-source precursor. As a result, the grown pillars show a 20–70 nm wide central iron-rich core surrounded by a shell with a Co:Fe ratio of 3:1. The high purity of the as-grown material is reflected in the unusually high crystallinity of the FEBID material. While the crystallite sizes can be increased by EBC, the data from associated FFT images do not allow unambiguous assignment of specific crystalline phases.

The present study does not question or challenge the current understanding of the FEBID process. The results described herein will contribute to the general understanding of the FEBID process under certain conditions. Typically, the dominating contributions are considered to be a fragmentation dominated by low-energy secondary electrons. In addition, the results presented provide important information that high energy channels are important contributors to apex formation in pillar growth. The apex should not form without contributions from the higher energy electrons under certain boundary conditions, such as working in the precursor limited regime and static beam conditions, etc. To the best of our knowledge, different regimes with drastically altered composition have not been encountered in the FEBID literature, as deposits based on the monometallic precursors may differ only slightly in composition or microstructure.

## Data Availability

All data relevant to this article are included in the figures and/or in the discussion.

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
