# Peer review of "Pillar Growth by Focused Electron Beam-Induced Deposition Using a Bimetallic Precursor as Model System: High-Energy Fragmentation vs. Low-Energy Decomposition"

_nanomaterials, 2023, doi:10.3390/nano13212907_

Round 1
Reviewer 1 Report
Dear Editor,
This is an interesting article on electron-beam induced pillar growth, which I recommend for publication, since it reveals details of the nanostructure growth mechanism. There are a few typo-like instances which I would like the authors to evaluate.
- The axes titles of Fig. 2(a) contain “NW,” which should denote “nanowire, however, this abbreviation has not been defined in the text.
- I was confused for a while with Fig. 2(a) trying to make sense of labels “z” and “y” with actual geometric axis labels, until I realized they are just arbitrary labels indicating position along the nanowire length. Concentrating on Fig. 2(a), before seeing Fig. 2(b) maybe contributed to my confusion. Using other letters as labels might avoid similar confusion for other readers.
- The y axes in the colored graphs have units of “at%,” but the numbers show atomic fractions, not percentages (a pure element should be 100 at%).
- The letters in the caption for Fig. 3, corresponding to Figs. 3(d) and (e) are mislabeled.
- Line 217 has a remaining parenthesis after EBC.
Few instances of the wrong use of the article "the," however, the meaning was not altered by this minor grammatical error.
Reviewer 2 Report
1. Please elaborate on the specific electron-induced fragmentation mechanisms involved in the decomposition of the HFeCo3(CO)12 precursor during the focused electron beam induced deposition (FEBID) process?
2. The authors have mentioned that metallic contents of up to >95 at% were achieved in the fabricated nanostructures. How does the choice of precursor and deposition parameters contribute to such high metal content?
3. Please provide more details on the microstructure and composition analysis techniques used in this study, especially the insights gained from energy-dispersive X-ray spectroscopy (EDX) and transmission electron microscopy (TEM)?
4. The conclusion highlights the observation of different fragmentation channels during the FEBID process using single-source bimetallic precursors. Please explain how these observations can be practically leveraged for controlled nanostructure growth?
5. In the context of nanopillar growth, please further explain the differences between high-energy fragmentation channels and low-energy decomposition regimes, and how they influence the resulting structures?
6. How do the findings regarding altered decomposition paths when high-energy electrons interact with the precursor contribute to our understanding of FEBID mechanisms?
7. The conclusion mentions a specific core-shell structure with a Co:Fe ratio of 3:1. Could you provide insights into how this composition affects the properties and applications of the grown nanopillars?
8. Are there any potential practical applications or technological advancements that can be derived from the knowledge of these distinct fragmentation channels in FEBID processes?

Round 2
Reviewer 2 Report
The paper can be accepted for publication